# CACE-Net: Co-guidance Attention and Contrastive Enhancement for Effective Audio-Visual Event Localization

Xiang He*
Xiangxi Liu*
Yang Li*
Brain-inspired Cognitive Intelligence Lab,
Institute of Automation, Chinese Academy of Sciences
Beijing, China
{hexiang2021,liuxiangxi2024,liyang2019}@ia.ac.cn

Dongcheng Zhao
Brain-inspired Cognitive Intelligence Lab,
Institute of Automation, Chinese Academy of Sciences
Beijing, China
Center for Long-term Artificial Intelligence
Beijing, China
zhaodongcheng2016@ia.ac.cn

Guobin Shen
Brain-inspired Cognitive Intelligence Lab,
Institute of Automation, Chinese Academy of Sciences
Beijing, China
Center for Long-term Artificial Intelligence
Beijing, China
shenguobin2021@ia.ac.cn

Qingqun Kong†
Brain-inspired Cognitive Intelligence Lab,
Institute of Automation, Chinese Academy of Sciences
Beijing, China
qingqun.kong@ia.ac.cn

Xin Yang†
Institute of Automation, Chinese Academy of Sciences
Beijing, China
xin.yang@ia.ac.cn

Yi Zeng†
Brain-inspired Cognitive Intelligence Lab,
Institute of Automation, Chinese Academy of Sciences
Beijing, China
Center for Long-term Artificial Intelligence
Beijing, China
Key Laboratory of Brain Cognition and Brain-inspired
Intelligence Technology, CAS
Shanghai, China
yi.zeng@ia.ac.cn

## Abstract

The audio-visual event localization task requires identifying concurrent visual and auditory events from unconstrained videos within a model, locating them, and classifying their category. The efficient extraction and integration of audio and visual modal information have always been challenging in this field. In this paper, we introduce CACE-Net, which differs from most existing methods that solely use audio signals to guide visual information. We propose an audio-visual co-guidance attention mechanism that allows for adaptive bi-directional cross-modal attentional guidance between audio and visual clues, thus reducing inconsistencies between modalities. Moreover, we have observed that existing methods have difficulty distinguishing between similar background and event and lack the fine-grained features for event classification. Consequently, we employ background-event contrast enhancement to increase the discrimination of fused features and fine-tuned pre-trained model to extract more discernible features from complex multimodal inputs. Experiments on the AVE dataset demonstrate that CACE-Net sets a new benchmark in the audio-visual event localization task, proving the effectiveness of our proposed methods in handling complex multimodal learning and event localization in unconstrained videos. Code is available at https://github.com/Brain-Cog-Lab/CACE-Net.

*Authors contributed equally.
†Corresponding author.

## CCS Concepts

• **Computing methodologies** → Object recognition.

## Keywords

Audio-Visual Event Localization, Audio-Visual Co-guidance Attention, Contrastive Enhancement

**ACM Reference Format:**
Xiang He, Xiangxi Liu, Yang Li, Dongcheng Zhao, Guobin Shen, Qingqun Kong, Xin Yang, and Yi Zeng. 2024. CACE-Net: Co-guidance Attention and Contrastive Enhancement for Effective Audio-Visual Event Localization. In *Proceedings of the 32nd ACM International Conference on Multimedia (MM '24), October 28-November 1, 2024, Melbourne, VIC, Australia.* ACM, New York, NY, USA, 9 pages. https://doi.org/10.1145/3664647.3681503

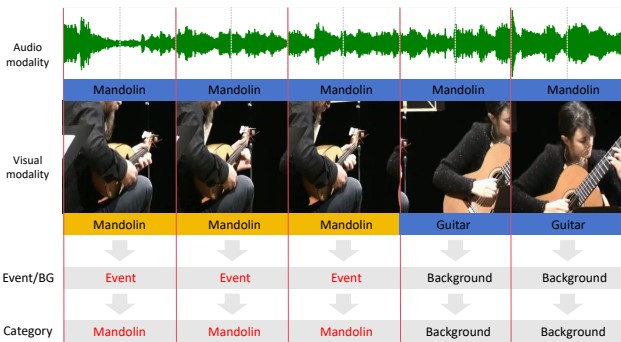

**Figure 1: An example of audio-visual event localization task, where we can identify an event as a mandolin only if both the sound and visual content are mandolin in the video segment.**

## 1 Introduction

Our brains perceive concepts and guide behavior by integrating clues from multiple senses, a process that relies on complex and efficient information processing mechanisms [3, 22]. This multisensory integration not only improves the robustness of perception but also its efficiency. Just as the brain enhances comprehension and decision-making through clues from multiple senses, multimodal learning also integrates information across different sensory modalities to achieve a more accurate and comprehensive understanding of the context. Especially in complex tasks, multimodal contextual comprehension is crucial for the current task and future predictions [25]. The task of audio-visual event localization [28] is a specific demonstration of this concept, which requires the identification of concurrent visual and auditory events from unconstrained videos rich in visual images and audio signals. As shown in Figure 1, for an unconstrained video, the audio-visual event exists only if the video's audio and visual information is matched in the video segment; all other cases are considered as background. Compared to action recognition that relies solely on visual information, audio-visual event localization requires a higher level of context analysis and comprehension due to the interference of asynchronous visual and audio signals, thereby increasing the complexity of recognition.

In recent years, researchers have proposed various methods to improve the performance of this task. For example, AVEL [28] utilizes a cross-modal attention mechanism to guide the processing of complex visual information with audio signals; CMRAN [31] further explores information within and across modalities. Despite significant progress, we argue that the task still faces three key challenges: 1) Interference in audio signals: the presence of noise or weak signals in audio can interfere with the process of guiding visual modality and with the generation of fused features. 2) Difficulty in distinguishing between background and event: The logic underlying event classification and distinction between background and event differs. The definition of "mismatch between audio and visual clues" results in a lack of distinct features for background, leading to challenges. 3) The need for fine-grained features: The main reason for misclassification of different events is the lack of fine-grained features, especially when events resemble one another.

To mitigate the above challenges, method design should follow the following principles: 1) Balance multimodal input: Reduce noise interference in the audio signal to avoid its excessive impact on feature fusion and event localization. 2) Fine-grained feature representation: Precisely capture background details and discriminate the subtle differences between the event and the background from the complex multimodal inputs. 3) Stable and efficient feature encoding: Encoders need to have strong generalization capability for feature extraction to achieve event classification more accurately.

In this paper, we present an audio-visual co-guidance attention mechanism. Unlike the previous approach of using only audio signals to guide visual modality, visual information is also used to guide audio modality. The co-guidance mechanism provides different information for event localization based on visual and audio modalities, and it allows the visual and audio modalities to obtain guidance signals from each other. The audio-visual co-attention mechanism reduces the impact of misleading information on audio-visual event localization by reducing the inconsistency of inter-modal information. Additionally, to address the challenge of distinguishing between background and event, we integrate a contrastive learning strategy into our model. This strategy involves intentionally perturbing features extracted by the model to simulate disturbance in scenes, which enhances the model's comprehension of background and event features, and focuses the learning process on distinguishing event from background containing information relevant to event more effectively. Finally, we fine-tune efficient visual and audio encoders specifically to extract more refined and discriminative features from complex multimodal inputs, thus enhancing the model's capability for generalized feature extraction. These three coordinated strategies are combined to propose a novel audio-visual event localization model that surpasses existing state-of-the-art methods in handling audio-visual event localization tasks.

Our contribution can be summarized as follows:

- Co-guidance attention: we introduce a novel audio-visual co-guidance attention mechanism, which effectively reduces the inconsistency of inter-modal information and the influence of misleading information on event localization through bi-directional guidance between visual and audio modalities.
- Background and event contrast enhancement: we use a contrastive learning strategy by randomly perturbing the fused features, e.g., adding noise, to enhance the contrast between event and background, and deepen the model's comprehension of background and event features.
- A more efficient feature extractor: we choose more advanced visual and audio encoders, and perform targeted fine-tuning in the audio-visual event localization task to extract fine-grained features from complex multimodal inputs.

## 2 Related Work

We begin with an introduction to the work on audio-visual learning, followed by a discussion of the attention mechanism and fine-tuning of the feature extractor on audio-visual event localization.

**Audio-visual learning.** The main goal of audio-visual learning is to mine the relationship between audio and visual modalities, in which feature fusion is the core research direction. The research ideas are mainly divided into two categories, one is to

use supervisory signals to complete the fusion of visual and audio information [12, 19], or to use one modality as a supervisory signal to drive the information mining of the other modality [1, 23]. The other focuses on cross-modal learning using unsupervised methods when there is a known correspondence between audio and visual modalities [2, 13, 20]. Development in the field of audio-visual learning has also led to an increasing number of tasks, including video sound separation [5], video sound source localization [14], action recognition [6], and audio-visual event localization [18, 28], etc. This paper focuses on the audio-visual event localization task.

**Attention mechanisms.** The attention mechanism on audio-visual event localization task was first proposed by [28], which explored the spatial correspondence of audio and visual modalities by using audio modality to guide the visual modality to implement a cross-modal attention mechanism. Later Wu et al. [29] proposed a dual-attention module to obtain the similarity of the two modalities. In addition, Xu et al. [31] proposed the module to guide the visual modality using audio information at spatial and channel levels to further explore the relationship between audio and visual modalities. However, the above work ignores the help of visual modality in guiding audio signals further. Feng et al. [4] use bi-directional guidance for closer audio-visual correlation. Our work improves on the above cross-modal attention mechanisms and can further reduce the inconsistency of information between audio and visual modalities. Unlike the introduction of more complex network module in CSS [4], we simply use the basic feature information for visual and audio bi-directional guidance to minimize the interference caused by noise in audio and visual signal when fusing features.

**Fine-tuning of the feature extractor.** In order to improve the suitability of features extracted by audio and visual encoders for downstream tasks, it is often chosen to fine-tune the feature extractor that has been pre-trained with a large-scale dataset using the dataset of the downstream task. Contrastive learning is considered as a very good way of fine-tuning. Training using contrastive learning is usually unsupervised, and the features obtained after training can be efficiently adapted to downstream tasks [9]. Radford et al. [24] proposed the CLIP framework, which is a contrastive learning method for textual and visual modalities. Guzhov et al. [8] adds audio modalities to the CLIP framework to construct the AudioCLIP framework. One of the central determinants of how well contrastive learning works is the definition of positive and negative sample pairs, and our work redefines positive and negative sample pairs to be more consistent with the event localization task.

## 3 Method

In this section, we first introduce the problem definition of the audio-visual event localization task. Subsequently, we present our proposed network framework, as depicted in Figure 2, which consists of three parts: audio-visual co-guidance attention, background-event contrast enhancement, and modal feature fine-tuning.

### 3.1 Problem Definition

We divide a given video sequence $\mathcal{S}$ into $T$ non-overlapping segments of one second each, denoted as $s_t$, where $\mathcal{S} = \{s_1, s_2, \ldots s_T\}$. Similarly, the corresponding visual and audio sequences can be represented as $\mathcal{V} = \{v_1, v_2, \ldots v_T\}$ and $\mathcal{A} = \{a_1, a_2, \ldots a_T\}$, respectively. The goal of audio-visual event localization is to accurately identify and locate event in those video segments where both visual and audio signals correspond to the same category. It requires the model to not only recognize whether the visual and audio inputs are matched but also accurately predicts the category of event.

Specifically, the model needs to predict the category of event associated with the visual segment $v_t$ and audio segment $a_t$ for each video segment $s_t$ within the input video sequence, where $t \in (1, T]$. Conversely, if the visual and audio does not match, meaning they belong to different categories of event or one of them does not represent any event category, the segment is considered as background. During training, we have access to segment-level labels, denoted as $\mathbf{y}_t$ within $s^t$, which indicate whether the video segment is considered as audio-visual event and which category it belongs to. Concretely, $\mathbf{y}_t = \left\{ y_t^c \mid y_t^c \in 0, 1, c = 1, \ldots, C, \sum_{c=1}^{C} y_t^c = 1 \right\}$, where $C$ is the number of categories. Here, $y_t^c = 1$ indicates the presence of event in $s_t$ and its category as $c$, while $y_t^c = 0$ indicates a mismatch of audio and visual information, which is regarded as background.

### 3.2 Audio-visual Co-guidance Attention

Although audio or visual information alone is not always reliable, they can provide each other with instructive information. Therefore, we use audio information to guide the processing of visual signals, which helps extract useful visual information for event localization from complex scene; similarly, we also use visual signals to guide audio processing to reduce the interference of noise mixed in the audio. As shown in Figure 3, through this audio-visual co-guidance attention (AVCA), the model can more robustly extract audio and visual features directly related to the event. For ease of representation, we denote the visual and audio features of the video segment $s_t$ as $v_t \in \mathbb{R}^{d_v \times (H*W)}$ and $a_t \in \mathbb{R}^{d_a}$, respectively.

**Audio-guided spatial-channel visual feature.** The work of researchers such as [28, 31] has demonstrated the importance of audio-guided visual features. Following Xu et al. [31], we obtain audio-guided channel-spatial attentive visual features $v_t^{cs} \in \mathbb{R}^{d_v}$. However, the reliability of these guided visual features is limited by the quality of the audio signal. Noise in the audio can obstruct the extraction of visual features related to audio-visual event.

Consequently, to robustly extract visual regions related to event, inspired by [15, 30], we utilize global features $v_t^g \in \mathbb{R}^{d_v \times (H*W)}$, obtained from spatial self-attention of the visual signals, to generate channel-level calibration features $\tilde{v}_t^g \in \mathbb{R}^{d_v}$. These calibration features are used to adaptively modulate the audio-guided visual features, selectively emphasizing useful features while suppressing less useful ones. The calibration features $\tilde{v}_t^g$ can be described as:

$$
\begin{aligned}
\tilde{v}_t^g &= \text{Softmax}\left(\mathbf{U}_v v_t^{gc}\right) \otimes (v_t^g)^T, \\
v_t^{gc} &= \mathbf{W}_1 F_{sq}(v_t^g) \odot \mathbf{W}_2 v_t^g
\end{aligned}
\tag{1}
$$

where $\otimes$ represents matrix multiplication, $\odot$ represents element-wise multiplication, $\mathbf{W} \in \mathbb{R}^{d \times d_v}$ is a fully connected layer with ReLU activation function, and $d$ represents the dimension of the hidden layer. $\mathbf{U}_v \in \mathbb{R}^{1 \times d}$ is a fully connected layer with a Tanh activation function. $F_{sq}$ represents the operation of compressing global spatial information into channel representations, defined

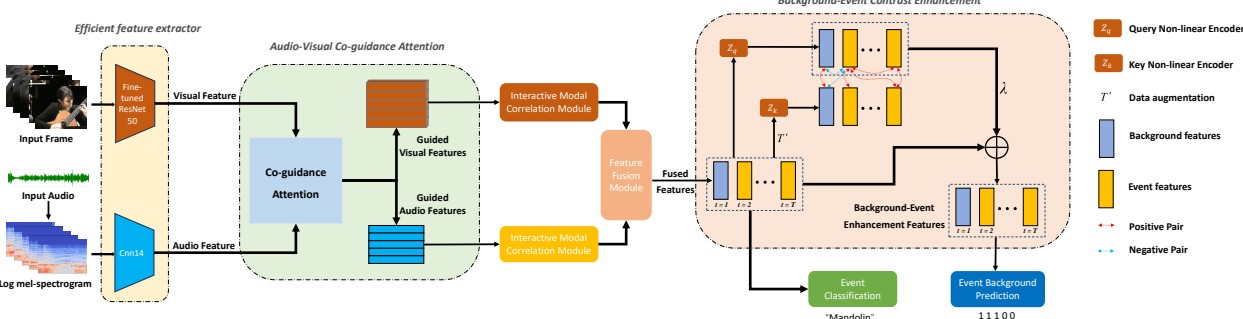

**Figure 2: Overview of our proposed network framework, which consists of three parts: audio-visual co-guidance, background-event contrastive learning, and pre-trained model targeted fine-tuning.**

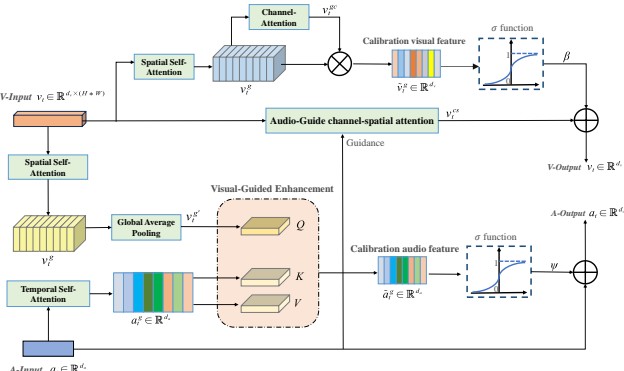

**Figure 3: Schematic diagram of audio-visual co-guidance attention. Based on visual and audio modalities providing different information for event localization, visual and audio acquire guidance signals from each other and adaptively conduct cross-modal attentional guidance.**

as $F_{sq}(v_t^g) = \frac{1}{H \times W} \sum_{i=1}^H \sum_{j=1}^W v_t^g(i, j)$. After obtaining the global spatial visual feature $v_t^{gc} \in \mathbb{R}^{d \times (H*W)}$, the spatial attention scores are computed through the nonlinear layer $\mathbf{U}_v$ and then multiplied with the global information to generate channel-level features $\tilde{v}_t^g$. The visual features with audio guidance can then be represented:

$$v_t = v_t^{cs} + \beta \cdot \sigma(\tilde{v}_t^g) v_t^{cs} \tag{2}$$

where $\sigma$ represents the sigmoid function, and $\beta$ is a hyperparameter.

**Visual-guided enhancement audio feature.** Due to inherent noise in audio signals, it is necessary to guide audio features with visual information to reduce audio interference. To extract visual-related features from audio modality, we first employ self-attention on visual information to capture global features $\mathbf{v}^g \in \mathbb{R}^{T \times d_v \times (H*W)}$; unlike spatial-level self-attention for visual modality, we apply temporal self-attention to the audio input, meaning that current time features depend on other moments to derive the global audio features $\mathbf{a}^g$, which can be expressed as:

$$\mathbf{a}^g = \text{Softmax}\left(qK^T\right) V \tag{3}$$

where $q, K, V$ can be denoted as:

$$q = \mathbf{a}W^Q, K = \mathbf{a}W^K, V = \mathbf{a}W^V \tag{4}$$

where $\mathbf{a} \in \mathbb{R}^{T \times d_a}$ represents the audio inputs of video segments. $W^Q, W^K, W^V \in \mathbb{R}^{d_a \times d_a}$ are fully connected layers.

With the global spatial visual features $\mathbf{v}^g$ and the temporal global audio features $\mathbf{a}^g$, we squeeze the spatial visual features into channel representations, denoted as $\mathbf{v}^{g\prime} \in \mathbb{R}^{T \times d_v}$. This process effectively encapsulates spatial information within channel dimensions. The squeeze is achieved through the function $\mathbf{v}^{g\prime} = F_{sq}(\mathbf{v}^g)$. Following this, $\mathbf{v}^{g\prime}$ is used as a query input to extract audio features related to visual modality, which results in visual-guided audio features $\tilde{\mathbf{a}}^g$ for calibration. It adaptively acquires features from the audio features beneficial for event localization, and suppresses out the features that are not relevant to the event. The audio features $\tilde{\mathbf{a}}^g$ involved in calibration can be calculated according to Equation 3, and corresponding $q, K, V$ can be expressed as:

$$q = \mathbf{v}^{g\prime}W_v^Q, K = \mathbf{a}^g W_a^K, V = \mathbf{a}^g W_a^V \tag{5}$$

where $W_v^Q \in \mathbb{R}^{d_v \times d_a}, W_a^K, W_a^V \in \mathbb{R}^{d_a \times d_a}$ denote linear projection for dimensional alignment.

The audio output after visual guidance is:

$$a_t = a_t + \psi \cdot \sigma(\tilde{a}_t^g) a_t. \tag{6}$$

where $\sigma$ represents the sigmoid function, and $\psi$ is a hyperparameter. The $\tilde{a}_t^g$ denotes the value of $\tilde{\mathbf{a}}^g$ at time $t$.

### 3.3 Background-Event Contrast Enhancement

After passing through the bi-directional guidance, the audio-visual segments acquire event-related visual representation $v_t$ and audio representation $a_t$. To perform effective audio-visual event localization, comprehensive feature fusion of the audio-visual information is necessary. We utilize the method described by Xu et al. [31], starting with multi-head attention mechanism and residual connections to extract information within each modality. Then, we employ cross-modal relation attention mechanism. Here, features of a single modality are used as the query, while features concatenated across audio-visual dimensions serve as the key and value, which enables cross-modal information fusion without overlooking details within each modality. The feature fusion $\mathcal{F}_{av}$ is then obtained through an audio-visual interaction module.

To refine the distinction between background and event and acquire features with distinct background-event differentiation, we utilize background-event contrast enhancement (BECE). Concretely, we choose to enhance the fused features using supervised contrastive learning [16]. Specifically, the supervised contrastive learning is conducted within each video sample, where one group of samples for supervised contrastive learning comprises a video along with all its audio-visual segments that have undergone data augmentation. We choose to add Gaussian noise as the method for data augmentation. Each video contains $T$ audio-visual pairs, thus a training set consists of $2T$ samples. Since the categories of audio-visual event are the same across different segments of the same video, we define positive sample pairs by this way: audio-visual pairs labeled as background and their own augmented ones are positive sample pairs; audio-visual pairs labeled as event and all other audio-visual pairs labeled as event are positive sample pairs.

To prevent overfitting, we use only one nonlinear layer to optimize the fused features, serving both as the query and key encoder. The loss function for supervised contrastive learning can be expressed as follows:

$$\mathcal{L}^{contrast} = \frac{1}{2T} \sum_{i \in I} \mathcal{L}_i^{contrast}$$

$$= \frac{1}{2T} \sum_{i \in I} -\log \left\{ \frac{1}{|K(i)|} \sum_{k \in K(i)} \frac{\exp\left(f_i \cdot f_k / \tau\right)}{\sum_{p \in P(i)} \exp\left(f_i \cdot f_p / \tau\right)} \right\}, \quad (7)$$

where $I$ represents a set of samples for contrastive learning, where $i \in I \equiv \{1, 2, \ldots, 2T\}$ indexes the samples. $K(i)$ denotes the set of positive samples corresponding to the sample indexed by $i$, and $|K(i)|$ is the number of positive samples. $P(i) \equiv I \setminus i$ represents the set of samples excluding the one indexed by $i$. $f_i$ represents fused feature corresponding to the sample with index $i$, and $\tau \in \mathcal{R}^+$ is the temperature coefficient. The loss function $\mathcal{L}^{contrast}$ enhances the optimization within the feature space, promoting the clustering of features labeled as event closer together, while distancing them from the features labeled as background. It encourages more discriminative feature representations for accurate event localization.

To preserve the information in the original fused features, we adopt the following method to derive the fused features:

$$\mathcal{F}_o = \mathcal{F}_{av} + \lambda \mathcal{F}_{FT} \quad (8)$$

where $\mathcal{F}_o$ represents the final fused feature used for binary classification between events and background. $\mathcal{F}_{av}$ is the original fused feature, and $\mathcal{F}_{FT}$ is the feature enhanced through supervised contrastive learning. This approach allows for preserving the original feature information while incorporating optimized features obtained through supervised contrastive learning, effectively balancing the model's generalization ability and localization accuracy.

### 3.4 Modal Feature Fine-tuning

We utilize visual and audio encoders, specifically ResNet-50 [10] and Cnn14 [17], which are pre-trained on large-scale datasets ImageNet and AudioSet respectively. They allow for efficient extraction of image and audio features. Given that the Audio-Visual Event (AVE) dataset is a subset of AudioSet [7], we fine-tune the visual encoder using contrastive learning on this dataset to better adopt

to audio-visual event localization tasks. We do average pooling of all the extracted features obtained from one video as samples in the contrastive learning dataset. This process provides each video with a set of visual and audio samples. Visual and audio samples from the same video serve as positive pairs, while samples from different videos serve as negative pairs.

The loss function used for contrastive learning is $L_{InfoNCE}$, defined as follows:

$$L_{InfoNCE} = -\frac{1}{B} \sum_{i=1}^{B} \log \frac{\exp\left(f_i^I \cdot f_i^A / \tau\right)}{\sum\limits_{j=1}^{B} \exp\left(f_i^I \cdot f_j^A / \tau\right)} \quad (9)$$

Here, $B$ represents the number of samples in a contrastive learning batch, $f^I$ and $f^A$ denote visual and audio features respectively, and $i$ and $j$ are index of visual and audio samples. The temperature parameter $\tau \in \mathcal{R}^+$ influences the separation of the feature space. This loss function ensures that during backpropagation, features of positive pairs become more similar, while features of negative pairs diverge, thereby enhancing the generality of the features.

### 3.5 Classification and Objective Function

For the final task of audio-visual event localization, we divide it into two subtasks. The first is the model's ability to detect whether the visual and audio information is matched, and the second is the model's ability to accurately determine the category of the event on a video-level basis. For the task of determining whether a video segment is background or event, we use the contrast-enhanced fused feature $\mathcal{F}_o \in \mathbb{R}^{T \times d}$. This allows us to obtain the probability scores of event occurrence $\hat{y} = \{\hat{y}_1, \hat{y}_2, \cdots, \hat{y}_T\}$:

$$\hat{y} = \sigma\left(W_3 \mathcal{F}_o\right), \quad (10)$$

where $\sigma$ denotes the sigmoid function, and $W_3 \in \mathbb{R}^{1 \times d}$ represents a fully connected layer with a single output neuron.

For classifying the category of audio-visual event, we use the original fused feature $\mathcal{F}_{av} \in \mathbb{R}^{T \times d}$. Further, after obtaining video-level feature through max pooling across the temporal dimension, we connect it to a classification linear layer $W_4 \in \mathbb{R}^{d \times C}$. This setup yields the video-level event category scores $\hat{y}c$ as follows:

$$\hat{y}_c = \text{Softmax}\left(W_4 \max(\mathcal{F}_{av})\right), \quad (11)$$

where $\max(\mathcal{F}_{av})$ denotes the max pooling operation across the time dimension, extracting the most prominent features for each category, and Softmax is applied to convert the linear layer outputs into probabilities for each event category.

During the training phase, since we have access to the event labels $y_t = \{y_t^c \mid y_t^c \in 0, 1\}$, we use $y_t$ as the label for binary classification between background and event, and $y_c = \arg\max_c(y_t)$ as the label for event category classification to train the model.

Consequently, the loss function for training the model can be expressed as:

$$\mathcal{L} = \mathcal{L}^c + \frac{1}{N} \sum_{t=1}^{N} \left(\mathcal{L}_t^e + \mathcal{L}_t^{sup}\right) + \mathcal{L}^{contrast}, \quad (12)$$

where $\mathcal{L}^c$ represents the cross-entropy loss for event category classification, $\mathcal{L}_t^e$ represents the cross-entropy loss for background-event classification, and $\mathcal{L}^{contrast}$ represents the contrastive learning

loss. Note that we incorporate the loss $\mathcal{L}_t^{sup}$ used for background suppression from [30] to suppress asynchronous audio-visual background within events and enhance audio-visual consistency.

During the inference phase, the model's prediction for each video segment $s_t$, denoted as $o_t$, is determined by both $\hat{y}_t$ and $\hat{y}_c$. The prediction is computed as $o_t = H(\hat{y}_t - \theta) * \hat{y}_c$, where $H$ is the Heaviside step function, and $\theta$ is a threshold for determining the presence of the event. In this work, we set this threshold $\theta$ to 0.5.

## 4  Experiments

In this section, we first discuss the experimental setup and then provide a series of ablation studies to show the effectiveness of the various components of the proposed method. Finally, we conduct experiments on the audio-visual event (AVE) dataset to compare our method with current state-of-the-art methods.

**Audio-visual event dataset.** Consistent with existing work [28, 30–32], we evaluated our approach on the publicly available AVE dataset. The AVE dataset [28] is a subset of the AudioSet [7], which contains 4,143 videos with a total of 28 event categories and 1 background category. The AVE dataset covers videos of a variety of realistic scenarios such as musical instruments playing, male speaking and train whistle, etc. Each video has a duration of 10 seconds and contains at least one event category. Each video is divided into 10 segments, each lasting 1 second, with labels assigned to each segment. Each video has at least 2 segments labeled as audio-visual event. In the audio-visual event localization task, the method we propose needs to predict for each video segment whether it is background or event and its event category.

**Implementation details.** For feature extraction, we explored on two types of encoders. One is the configuration in the original work [28], where visual features are averaged using the "pool5" layer of VGGNet-19 [27] pre-trained on ImageNet [26] for 16-frame-per-second in video, and audio features are extracted using the VGGish [11] pre-trained on AudioSet, which converts the audio to the log-spectrogram. The other is the ResNet-50 [10] pre-trained on ImageNet and the Cnn14 network [17] pre-trained on AudioSet. The latter is fine-tuned on the AVE dataset through contrastive learning to obtain more generalized feature representations.

### 4.1  Ablation Study

In this section, we conduct experiments to validate the effectiveness of each part of the proposed method. In subsequent experiments, we uniformly use visual features extracted by VGG-19 with audio features extracted by VGGish as model inputs.

**Effectiveness of audio-visual co-guidance attention.** In our approach, we implement a bi-directional guidance mechanism based on the fact that visual and audio modalities provide different information for event localization and they access each other's guidance signals to collaborate in cross-modal attentional guidance. Therefore, we compare audio-visual co-guidance attention with three other attentional guidance mechanisms: 1) "w/o AVCA": where visual and audio features are directly fed into the network for integration without any guidance 2)"w/ AVCA-Visual-only": where audio information guide visual modality while preserving the original audio features 3) "w/ AVCA-Audio-only": where visual information guide audio modality while preserving the original visual features.

**Table 1: The ablation study of the audio-visual co-guidance attention mechanism. We compared AVCA with three other attentional guidance mechanisms.**

| Method | Accuracy (%) |
| --- | --- |
| w/o AVCA | 78.26 |
| w/ AVCA-Visual-only | 78.83 |
| w/ AVCA-Audio-only | 77.23 |
| w/ AVCA | **80.30** |

**Table 2: Ablation experiments for background-event contrast enhancement. Experiments are conducted on the basis of audio-visual co-guidance attention. *Represents results of contrast-enhanced fused features used for both event-background binary classification and event classification.**

| Methods | Accuracy (%) |
| --- | --- |
| w/o BECE | 80.30 |
| w/ BECE $\lambda = 0.2$ | 80.25 |
| w/ BECE $\lambda = 0.4$ | 80.65 |
| w/ BECE $\lambda = 0.6$ | **80.80** |
| w/ BECE $\lambda = 0.8$ | 80.10 |
| w/ BECE $\lambda = 0.6$ | 80.65* |

The experimental results are shown in Table 1, where it can be observed that the performance of the network decreases dramatically when the audio-visual co-guidance attention module is removed, which indicates that AVCA plays an important role in reducing the inconsistency of the inter-modal information and the impact of misinformation. Consistent with existing work, we find that audio-guided visual signals have a nearly 0.6% accuracy improvement for the network compared to no guidance, which suggests that audio signals can be used to allow visual learning to focus on event-related regions.

Surprisingly, the method w/ AVCA-Visual-only does not result in performance gain. On the contrary, its inclusion has a negative effect. We attribute the result to the significant background noise present in visual scenes. Directly guiding the audio modality using visual signals may result in the neglect of event-related information in the audio modality, and may even lead to focusing on audio noise instead. Finally, the improvement brought by the audio-visual co-guidance attention is significant. It gives a 2.04% accuracy improvement compared to not using the AVCA method, proving the superiority of the audio-visual co-guidance attention.

**Effectiveness of background-event contrast enhancement.** In our approach, we propose using supervised contrastive learning to optimize the fused feature $\mathcal{F}_o$ for classification between event and background. We first obtain the background-event contrast enhancement fused feature $\mathcal{F}_{FT}$ and merge it with the original fused feature $\mathcal{F}_{av}$ under a certain weight $\lambda$. We compare the results under different weights with those obtained without background-event contrast enhancement, as shown in Table 2. From the table, it can be observed that when the weight is appropriately selected ($\lambda = 0.4$ and $\lambda = 0.6$), the model achieves a accuracy improvement of 0.5%, which indicates that supervised contrastive learning effectively help the model distinguish between event and background. However,

**Table 3: Comparisons with encoders and fine-tuning.**

| Method | Accuracy (%) |
|---|---|
| w/o More Efficient Encoders | 80.80 |
| with More Efficient Encoders | 81.56 |
| with More Efficient Encoders + Fine-tuning | 82.36 |

**Table 4: Ablation experimental results overview.**

| AVCA | BECE | Efficient Encoder | Accuracy(%) |
|---|---|---|---|
| - | - | - | 78.83 |
|  | ✓ |  | 78.93 |
| ✓ |  |  | 80.30 |
| ✓ | ✓ |  | 80.80 |
| ✓ | ✓ | ✓ | **82.36** |

when the weight is too high ($\lambda = 0.8$), the model performance decreases, which suggests that the fine-tuned fused feature does not entirely reflect the visual modality information and also proves the necessity of retaining the original fused feature $\mathcal{F}_{av}$.

Furthermore, we test the performance of using the enhanced fused feature under the optimal parameters for both event background binary classification and event classification. The results show that although the model's accuracy increases from 80.30% to 80.65% after enhancement, it still lower than the 80.80% achieved when the enhanced fused features are used exclusively for event-background binary classification, indicating that the original features perform better for event classification. We attribute the result to the fact that since the supervisory signal for supervised contrastive learning is the event-background binary classification label, it serves to further distance the event and background features in the feature space, but does not separate the features of different events. Moreover, contrastive learning may even alter the relative positions of features for certain event in the feature space, reducing the distance between them and features of other event categories, thereby leading to a degradation in model performance.

**Effectiveness of a more efficient encoder.** In our approach, we employed ResNet50 and Cnn14 [17] as visual and audio feature extractors, respectively, which have been pre-trained on the ImageNet and AudioSet datasets. These encoders outperform the VGG19 and VGGish models previously used for extracting baseline features, providing more robust feature extraction which contributes to enhanced model performance. Additionally, we fine-tune the visual encoder, ResNet50, to tailor its capabilities more closely to the audio-visual event localization task. We compared these results with those obtained using the original encoder and switching to more efficient encoders, and the outcomes are presented in Table 3.

The results indicate that using more efficient encoders for feature extraction can increase model's accuracy by 0.8%. Furthermore, features extracted using the fine-tuned ResNet50 provided an additional 0.8% accuracy improvement. This not only demonstrates that using more efficient encoders for feature extraction perform better for downstream tasks but also validates the effectiveness of fine-tuning encoders with downstream task datasets.

**Table 5: Comparisons with state-of-the-arts in a supervised manner on AVE dataset**

| Method | Feature | Accuracy (%) |
|---|---|---|
| Audio [28] | VGG-like | 59.5 |
| Visual [28] | VGG-19 | 55.3 |
| AVEL [28] | VGG-like, VGG-19 | 72.7 |
| DAM [29] | VGG-like, VGG-19 | 74.5 |
| CMRAN [31] | VGG-like, VGG-19 | 77.4 |
| PSP [32] | VGG-like, VGG-19 | 77.8 |
| CMBS [30] | VGG-like, VGG-19 | 79.3 |
| AVE-CLIP [21] | VGG-like, VGG-19 | 79.3 |
| CSS [4] | VGG-like, VGG-19 | 80.5 |
| **CACE-Net (ours)** | VGG-like, VGG-19 | **80.8** |
| **CACE-Net (ours)** | Cnn14, Res-like | **82.4** |

**Summary of ablation experiment results.** To clearly see the contribution of each part of our proposed method to the model performance, we summarize the experimental results of the three methods, namely, audio-visual co-guidance attention (AVCA), background-event contrast enhancement (BECE), and more efficient encoder, in Table 4. It can be seen that AVCA, BECE each used individually gives an improvement in model performance. When they are used together, the accuracy of 80.80% is achieved. Compared to the baseline result, i.e., 78.83% accuracy achieved without any of our methods, the accuracy is improved by about 2%, proving the effectiveness of our methods. Further, by adding fine-tuned efficient encoders for feature extraction on AVCA and BECE, our method achieves the best result, i.e., 82.36%.

## 4.2 Comparison of State-of-the-Art methods

As shown in Table 5, our proposed CACE-Net model has a significant improvement over existing state-of-the-art methods. Previous methods, such as using only audio or visual unimodal information as model inputs in AVEL, had lower accuracy of 59.5% and 55.3%, respectively. Subsequent models, such as AVEL [28], DAM [29], and CMRAN [31], have integrated audio and visual modalities in more sophisticated manners, resulting in significantly improved accuracy. With the introduction of models such as PSP [32] and CMBS [30], the accuracy has improved to close to 80%. The CSS [4] model further refines this integration with the accuracy of 80.5%.

Our CACE-Net utilizes features extracted from VGG-like and VGG-19 to achieve the accuracy of 80.8%, outperforming all other methods. Additionally, we test the result of CACE-Net when extracting features using more robust encoders, achieving 82.4% accuracy.

## 4.3 Qualitative Analysis

We show the results of our method on two relatively difficult samples selected from the dataset, as shown in Figure 4. In the upper samples, the event category is labeled accordion; however, the audio signal is mixed with noise signals that are not relevant to the event, as well as signals that are relevant to other event categories, such as the "female speaking", which interferes with determining the event category. As a result, it is difficult to achieve good results without visual-guided audio features such as CMGA, which fails to predict the accordion event in the last three seconds. In

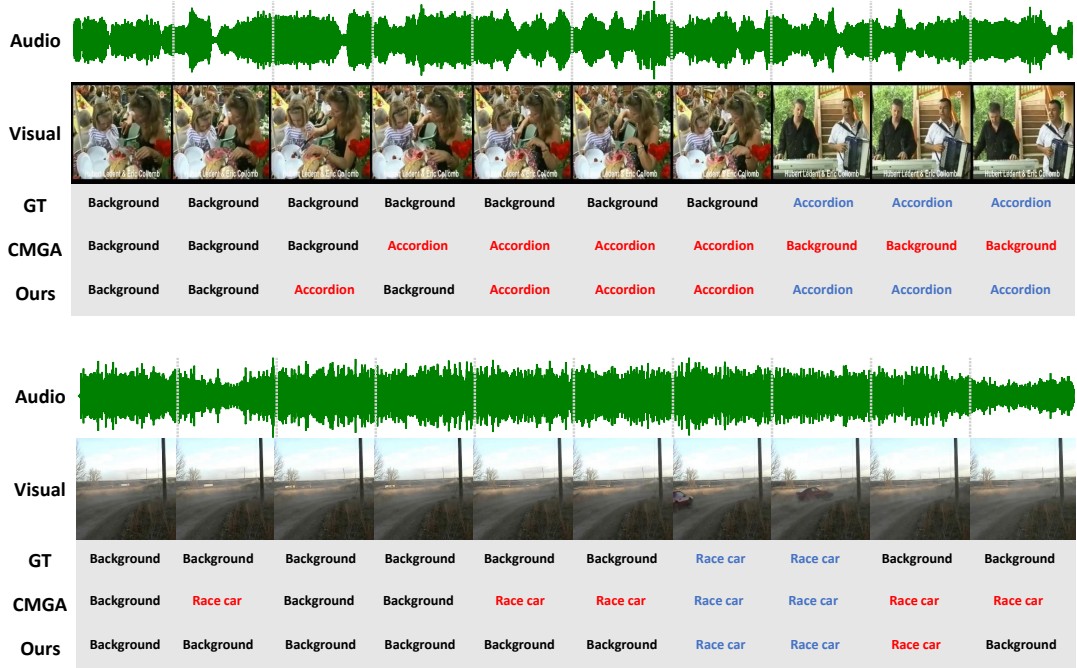

**Figure 4: Prediction results of our method on two relatively difficult samples selected from the AVE dataset. The black font represents the background, the blue font represents the categories of events that the model predicted correctly, and the red is the incorrect predictions output by the model. GT stands for Ground Truth.**

contrast to our method, CACE-Net achieves more accurate prediction results. However, the model still can't predict all correctly, i.e., incorrectly predicting the background as the accordion in 5-7 seconds. We attribute the experimental result to a highly dominant event-related audio signal in 5-7 seconds. Even though overall our method achieves superior discrimination, how this dominant signal can be further identified and optimized deserves further exploration, which we leave to subsequent work. In the sample below, the racing information only appears in the middle 2 seconds of the video, and our model correctly accomplishes this prediction as well.

In addition, we visualize different attentional guidance and the results are shown in Figure 5. The first three figures counting from the left show that our attentional guidance does not attend to non-event-related information. In the third figure, the event category is the horse, and attention is focused only on the horse itself and not on the man next to it. The last three figures counting from the left show that our attentional guidance is able to more accurately find information related to the event category, which suggests that our audio-visual co-guidance attention reduces inconsistencies in the audio-visual information and attends to the visual regions relevant to event more accurately.

## 5 Conclusion

In this paper, we propose an innovative network for audio-visual event localization by exploring multimodal learning to improve the comprehension and prediction of complex audio-visual scenes. Our model reduces the inconsistency of cross-modal information

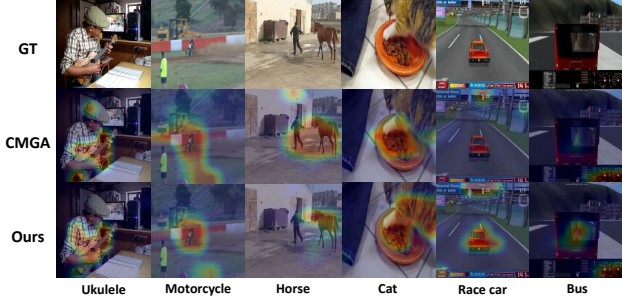

**Figure 5: Visualization of different attentional guidance methods. The top row is ground truth as the original image, the middle row is the CMGA method, and the last row is our method (CACE-Net).**

through the audio-visual co-guidance attention mechanism. In addition, we introduce a supervised contrastive learning strategy to improve the distinction between event and background by intentionally perturbing features, which further improves the model's ability to capture fine-grained features. Our experimental results on the task of audio-visual event localization show that CACE-Net can achieve better performance than existing state-of-the-art methods, validating the effectiveness of our proposed strategies in improving the generalization ability and classification accuracy of the models.

# 6 Acknowledgements

This research was financially supported by funding from the Institute of Automation, Chinese Academy of Sciences (Grant No. E411230101), and the National Natural Science Foundation of China (Grant No. 62372453).

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
