# OpenReview forum: "CACE-Net: Co-guidance Attention and Contrastive Enhancement for Effective Audio-Visual Event Localization"
_acmmm.org/ACMMM/2024/Conference — MM2024 Poster_

### Official Review · Reviewer_babS · 2024-05-14

**Rating:** 4
**Confidence:** 3

**Summary:**

In order to efficiently identify concurrent visual and auditory events from unconstrained videos, locate them, and classify their categories, this paper proposes CACE-Net, an innovative audio-visual event localization network where they design an audio-visual co-guidance attention mechanism and employ a background-event contrast enhancement (BECE) method.
This work has some significance about how to integrate audio and visual modal information and capture fine-grained features for event classification. But more experimental analyses need to be provided, such as considering the impact of using different data augmentation methods on the event-background prediction.

**Strengths:**

1. A novel audio-visual co-guidance attention mechanism is proposed for effectively reduces the inconsistency of inter-modal information.
2. A background-event contrast enhancement module that uses the contrast learning strategy to randomly perturb the fusion features is proposed for enhancing the contrast between the event segments and the background.
3. More advanced visual and audio encoders are chosen by performing targeted fine-tuning to extract fine features from complex multi-modal inputs.

**Limitations:**

1. The method of data augmentation is important in BCBE module. Can the authors give the results of using different data augmentation methods on the event-background prediction?
2. In Figure 2, it is better to add an arrow between background feature and the enhanced background feature, which can be consistent with the description of positive pairs in section 3.3.
3. To verify the effectiveness of the targeted fine-tuning encoders, adding a targeted fine-tuning experiments on the original encoders is necessary.
4. In Equation 12, the weights corresponding to different loss functions are the same. Will changing the weights to different values have an impact on the results?
5. The experiments are all conducted on the AVE datasets, and the authors can consider other similar datasets to evaluate the model's generalization ability.

**Suitability:**

3

---

### Official Review · Reviewer_UT6G · 2024-05-23

**Rating:** 3
**Confidence:** 2

**Summary:**

The paper introduces CACE-Net, a novel approach for audio-visual event localization. This task involves identifying and classifying events in videos by integrating both audio and visual signals. The key contributions include an audio-visual co-guidance attention mechanism and a background-event contrast enhancement strategy. The co-guidance mechanism allows bidirectional attention between audio and visual inputs, improving cross-modal consistency. The contrast enhancement strategy leverages contrastive learning to better distinguish events from backgrounds. The proposed method demonstrates state-of-the-art performance on the AVE dataset, setting a new benchmark for audio-visual event localization.

**Strengths:**

1. Novelty: The introduction of an audio-visual co-guidance attention mechanism is a significant innovation. Unlike previous methods that primarily use one modality to guide another, this bidirectional approach ensures a more balanced and effective integration of audio and visual information.
2. Technical Approach: The paper presents a well-structured methodology. The co-guidance attention reduces inconsistencies between modalities, while the contrast enhancement fine-tunes the model’s ability to distinguish events from backgrounds. These techniques are backed by rigorous mathematical formulations and justified through empirical results.
3. Adequate Evaluation: Extensive experiments on the AVE dataset validate the effectiveness of CACE-Net. The ablation studies are thorough, demonstrating the contribution of each component to the overall performance.
4. Clarity: The paper is well-written, with clear explanations of complex concepts. The figures and tables effectively illustrate the model architecture and experimental results.
5. Applications: The proposed method has significant implications for multimedia processing, particularly in scenarios requiring robust event localization in videos with varying backgrounds and noise levels.

**Limitations:**

1. Complexity: While the paper introduces novel mechanisms, the added complexity might be a challenge for practical implementations. The bidirectional attention mechanism and contrastive learning strategy could increase computational overhead.
2. Generalization: Although the method performs well on the AVE dataset, its generalization to other datasets and real-world scenarios is not fully explored. Additional experiments on diverse datasets would strengthen the claim of robustness.
3. Fine-grained Features: The paper mentions the difficulty in distinguishing between similar events and backgrounds. Despite the contrast enhancement strategy, there might still be cases where fine-grained feature extraction falls short, as noted in some qualitative analyses.
4. Comparative Baselines: The comparison with existing methods primarily focuses on performance metrics. A deeper analysis of how CACE-Net handles specific challenging cases compared to baseline methods would provide more insight.

**Suitability:**

3

---

### Official Review · Reviewer_UzkY · 2024-05-26

**Rating:** 4
**Confidence:** 3

**Summary:**

The paper presents CACE-Net, a cutting-edge network model designed for audio-visual event localization. The core innovations of CACE-Net include:
1. Audio-Visual Co-Guidance Attention Mechanism: This mechanism facilitates bidirectional guidance between audio and visual inputs, enhancing feature extraction by reducing inconsistencies between the modalities.
2. Background-Event Contrast Enhancement: Incorporates contrastive learning to better distinguish events from the background, leading to more precise and discernible features.
3. Efficient Feature Extractor Fine-Tuning: Utilizes advanced encoders for both audio and visual data, fine-tuned for improved feature extraction from complex multimodal inputs.

Experiments conducted on the AVE dataset demonstrate that CACE-Net outperforms existing state-of-the-art methods, establishing a new benchmark in audio-visual event localization.

**Strengths:**

1. Novelty: The co-guidance attention mechanism and contrastive enhancement are novel and effectively address key challenges in audio-visual event localization.
2. Technical Correctness: The methods are theoretically sound, with detailed algorithmic descriptions and mathematical formulations supporting their validity.
3. Evaluation: Extensive experiments on the AVE dataset showcase CACE-Net's superiority over current methods. Ablation studies provide clear evidence of the contribution of each component.
4. Clarity: The paper is well-structured and clearly written, making the methodology and results easy to understand.
5. Applications: The model has practical applications in areas such as content analysis, and multimedia retrieval.

**Limitations:**

1. Generalization: The model's generalizability to other datasets or real-world scenarios with varying types of audio-visual events is not thoroughly explored.
2. Complexity: The bidirectional guidance mechanism and contrastive learning introduce additional complexity, which may pose challenges for real-time applications.
3. Real-world Noises: The paper lacks detailed analysis on how well CACE-Net handles real-world noise and occlusions in visual data.
4. Error Analysis: A detailed error analysis is missing, which could provide insights into the model's failures and areas for improvement.

**Suitability:**

3

---

### Meta-Review · Area_Chair_MWub · 2024-07-02

**Recommendation:** Accept (Poster)
**Confidence:** 5

**Metareview:**

The initial ratings are 1 BR and 2 BA, and they are 1 WA and 2 BA after rebuttal. The reviewers all appreciated the proposed method about audio-visual co-guidance attention mechanism, and are satisfied with the authors' responses.
The response to reviewers should be added in the final version.
I agree with the reviewers and recommend an acceptance.